# Influence of Sociodemographic and Lifestyle Factors on Depression and Anxiety in Patients with Rheumatoid Arthritis in Saudi Arabia

**DOI:** 10.3390/ijerph22111625

**Published:** 2025-10-25

**Authors:** Leena R. Baghdadi, Mohammed K. Alhassan

**Affiliations:** 1Department of Family and Community Medicine, College of Medicine, King Saud University, Riyadh 11362, Saudi Arabia; 2Department of Family and Community Medicine, King Saud University Medical City, Riyadh 11461, Saudi Arabia; 3College of Medicine, King Saud University, Riyadh 11362, Saudi Arabia; mohammed.alhassan@charite.de

**Keywords:** rheumatoid arthritis, DMARDs, depression, anxiety, sociodemographic factors, lifestyle habits

## Abstract

Background/Objectives: Patients with rheumatoid arthritis (RA) frequently experience depression and anxiety, adversely affecting their quality of life. Addressing mental health in this population is a key public health priority. This study is aimed at examining the influence of sociodemographic and lifestyle factors on these conditions, which is essential for comprehensive patient care. Methods: A cross-sectional study was conducted at a regional hospital in Riyadh between March and November 2022, involving 213 patients diagnosed with RA. Depression and anxiety levels were assessed using the Hospital Anxiety and Depression Scale, and sociodemographic and lifestyle information was collected via questionnaires and from patients’ medical records. To identify significant associations, bivariate and regression statistical analyses were performed. Results: The prevalence rates of depression and anxiety were 16.4% and 23%, respectively. Significant associations were found between sociodemographic factors (age, marital status, education level, healthcare facility type) and depression and anxiety levels. Lifestyle factors, specifically sugar-sweetened drink consumption, were significantly associated with anxiety. Conclusions: Sociodemographic and nutritional factors play a crucial role in shaping the psychological well-being of patients with RA. These findings highlight the importance of adopting holistic care strategies that address both the physical and mental health needs of these patients. Enhancing access to preventive medicine and public health services in Saudi Arabia is essential for achieving this goal. Future research should continue to explore these associations to guide the development of targeted interventions.

## 1. Introduction

Rheumatoid arthritis (RA) is a chronic inflammatory disease affecting approximately 0.5–1% of the global population, leading to significant disability, reduced quality of life, and increased mortality [1]. Despite advances in pharmacological treatments, the global burden of RA continues to rise, with prevalence projected to increase further by 2050 [2,3]. The impact of RA extends beyond physical symptoms, as mental health comorbidities—particularly depression and anxiety—are notably more common in RA patients than in the general population. These psychological conditions are linked to worse disease outcomes, higher pain levels, and greater functional impairment, yet they are often underrecognized and undertreated, which can negatively affect treatment adherence and overall disease management [4,5,6].

A growing body of international research highlights that lifestyle and sociodemographic factors—such as younger age, female gender, lower education, and unemployment—are associated with higher rates of depression and anxiety in RA [7,8,9,10,11,12,13,14,15]. Disease-related factors, including higher disease activity, pain, and disability, further contribute to the risk of depression and anxiety [3,9]. Comorbid medical conditions and poor quality of life also play a significant role in increasing vulnerability to depression and anxiety among RA patients [9]. However, most of these studies have been conducted in Western or Asian populations, leaving a gap in understanding the interplay of these factors in other regions, such as the Middle East and North Africa.

In Saudi Arabia, depression and anxiety are a significant public health concern, with depression and anxiety prevalent in both the general population and among those with chronic diseases [16]. Recent studies have shown that depression and anxiety are highly prevalent among Saudi RA patients, with depression rates reaching up to 68% in some cohorts [17]. These mental health comorbidities are strongly associated with impaired health-related quality of life and daily functioning, and their impact is often compounded by sociodemographic and disease-related factors such as age, gender, marital status, rheumatoid factor positivity, and prolonged disease duration [1,3,4]. Despite the rising burden of RA in the region, few studies have specifically examined the relationship between sociodemographic or lifestyle factors and depression and anxiety in Saudi RA patients, and most available research has focused on patients receiving care in specialized healthcare settings [18,19]. Furthermore, cultural, religious, and social factors unique to the Saudi context may influence both the experience of RA and the manifestation of depression and anxiety symptoms, yet these aspects remain underexplored [6].

Although a few international studies have been conducted to examine the influence of sociodemographic and lifestyle factors on depression and anxiety [20,21], there is a notable lack of evidence specific to Saudi Arabia. This gap highlights the need for research tailored to the local context, considering the unique cultural, social, and healthcare system factors that may affect both RA and depression and anxiety health outcomes. Therefore, the aim of this study is to explore the association between depression and anxiety (as measured by the Hospital Anxiety and Depression Scale) and sociodemographic and lifestyle factors in patients with RA in an outpatient setting in Saudi Arabia.

## 2. Materials and Methods

### 2.1. Study Population Design

This investigation forms part of a broader research initiative examining how disease-modifying antirheumatic drugs (DMARDs) influence depression and anxiety levels in Saudi patients with RA. The study was conducted in accordance with Good Clinical Practice and the Declaration of Helsinki, with detailed methodology previously reported [22].

Between March and November 2022, a questionnaire-based cross-sectional study was carried out, enrolling 213 adults diagnosed with RA from the outpatient clinics at King Khalid University Hospital (KKUH), a major tertiary care center serving a diverse patient population. Eligible participants were identified using the hospital’s Electronic System for Integrated Health Information (eSiHi), which facilitated the application of a simple random sampling method. Each eligible patient was assigned a unique identifier, and a random number generator was used to select participants without bias. Those selected were invited to participate and provided written informed consent before completing an electronic questionnaire. Inclusion criteria required participants to be at least 18 years old, have a confirmed RA diagnosis, and be receiving either conventional or biologic DMARDs [22]. Exclusion criteria included a history of malignancy; use of high-dose methotrexate (≥20 mg/week); age under 18; diagnosis of any psychiatric disorder; current use of antidepressants, anxiolytics, or antipsychotic medications; and active corticosteroid use. Medication records were reviewed to ensure that no participant had used antipsychotic drugs or corticosteroids within the three months preceding the study period.

### 2.2. Sample Size Estimation

The estimated sample size for this study was calculated as n = 213 participants. This estimate was based on prior research indicating a prevalence of approximately 30% for depression and anxiety among patients with RA [23]. The calculation used a 5% margin of error and an 85% confidence level. Detailed methodology for the sample size determination is provided in our previous publication [22]. Initially, a minimum of 175 participants was required; an additional 20% (equivalent to 38 participants) was added to account for potential nonresponses and incomplete data. Consequently, the final sample size was set at 213 participants to ensure adequate statistical power.

### 2.3. Data Collection

A questionnaire in Arabic or English (Appendix A) was sent to the selected eligible participants via online links to their phone numbers or email addresses, which were obtained from the patients’ hospital database. Given the lower response rate associated with electronic surveys compared to face-to-face surveys [24], reminder messages were sent via WhatsApp every three days for approximately two weeks (4–5 reminders). An electronic informed consent form had to be signed electronically before participating in the study. Further data pertaining to medical history and medications were obtained through phone interviews and reviews of patients’ medical records on eSiHi. To ensure the clarity and phrasing quality of the questionnaire, a pilot study involving 20 participants was conducted; these results were not included in the main study.

The questionnaire had three main parts: demographic features, common chronic diseases, and the HADS-A and HADS-D (HADS Anxiety and Depression Scale) questionnaires in English and Arabic [25,26] (Appendix A). Details about the three parts of the questionnaire were provided in our previous study [22]; in summary, the questionnaire about the main outcome (depression and anxiety, i.e., high scores on the Hospital Anxiety and Depression Scale [HADS]) comprised fourteen questions, seven addressing anxiety and seven focusing on depression. Each query allowed for responses on a 4-point scale ranging from 0 (lowest) to 3 (highest). The cumulative scores for each section were then converted into a scoring model, enabling the classification of patients’ outcomes into three categories: normal (0–7), borderline abnormal (8–10), and abnormal (11–21). Patients with abnormal scores within the range of 11–21 in either the anxiety or depression sections were diagnosed with the respective condition.

### 2.4. Data Analysis

Data analysis was conducted using SPSS version 25.0 (SPSS, Chicago, IL, USA). Descriptive statistics, including means, standard deviations, frequencies, and percentages were employed to summarize quantitative and categorical variables. Bivariate statistical analyses were carried out using Student’s t-test and one-way analysis of variance. The Chi-square test was utilized to assess differences in depression and anxiety levels across patients based on sociodemographic factors, physical activity, dietary habits, and smoking status. Statistical significance was determined with *p*-values < 0.05 and 95% confidence intervals (CIs). Multiple linear regression analyses were performed to explore the associations between depression, anxiety scores, and sociodemographic and lifestyle factors, with the model adjusted for potential confounders such as age, gender, and disease activity (CRP and ESR levels).

## 3. Results

### 3.1. Sociodemographic Characteristics and Clinical History

As mentioned in the Methods section, the study population is from our previous publication [22]. There were 213 participants, as previously reported, and their sociodemographic characteristics are presented in Table 1 [22]. Of this RA population (n = 213), 45.1% (96) of the participants were between 51 and 65 years old. They were predominantly female, with 85.9% (183) women and 14.1% (30) men, resulting in a female-to-male ratio of 6.1 to 1. Most participants (93.9%) were Saudi nationals, with a significant proportion (40%) residing in the Northern region of Riyadh, and 68.1% were married. There were 7.5% illiterate participants and 51.6% of the participants held university degrees. Employment data indicated 67.6% of the participants were unemployed. Nearly 29.1% of participants earned less than 5000 Saudi riyals per month. Time to reach hospital from home was 16–30 min for 43.3% of participants, while 19.7% of participants required over 1 h. Despite 84.0% of the participants having health insurance, 88.7% of the study population accessed follow-up care at government health facilities.

The participants’ clinical and medication history was extracted from our previous publication [22] (Table A1). Approximately one-third of the patients with RA had been living with the condition for 6 to 10 years. The most recent levels of inflammatory markers (from eSIHI) were assessed, showing mean C-reactive protein (CRP) and erythrocyte sedimentation rate (ESR) levels of 8.6 ± 18.5 μg/mL and 41.7 ± 25.6 mm/h, respectively. The study population of RA patients was treated with various conventional and biologic DMARDs; methotrexate and hydroxychloroquine were the most frequently used conventional DMARDs, exclusively prescribed to 56 (26.3%) and 25 (11.7%) patients, respectively. In addition, around 20% of the patients were prescribed a biologic DMARD, adalimumab and etanercept being the most common. The analysis further indicated that 36.2% of participants were on a combination of RA medications.

### 3.2. Depression and Anxiety, and Sociodemographic Factors

Based on the HADS scores, 35 (16.4%) and 49 participants (23%) were classified as having depression and anxiety, respectively [22]. A comparison of depression and anxiety levels in patients with RA, based on their sociodemographic characteristics, is presented in Table 2. Sociodemographic factors of patients with RA, including gender, employment status, age, level of education, household income, number of family members, area of living, time between home and hospital, health insurance, and facility for receiving DMARDs injections showed no significant associations with the level of depression and anxiety (*p*-value > 0.05). However, the type of healthcare facility where RA patients had access to care was the only factor that showed a statistically significant difference (*p* = 0.048) in depression levels. The RA patients who visited private facilities (58.4%) had a significantly higher prevalence (*p*-value = 0.004) of borderline and abnormal depression compared with those who accessed government facilities (36.5%).

Table 3 shows the participants’ physical activities, dietary habits, and smoking status. There was no significant relationship between these characteristics and the level of depression and anxiety (*p*-value > 0.05), except for consumption of sugar-sweetened drinks. There is a link between consuming sugar-sweetened beverages and increased anxiety levels, with a notable number of patients in the abnormal anxiety category. The prevalence of RA patients who consumed sugar-sweetened drinks 4–6 times in a week showed significant abnormal levels of anxiety, while those who never consumed sugar-sweetened drinks had normal anxiety scores (38.5% vs. 68%, *p* = 0.038).

The test results in Table 4 show that in the unadjusted model, age, educational level, marital status, and the health facility visited have a significant effect on the patient’s anxiety score at a significance level of 5%, while household monthly income has a significant effect at a significance level of 10%. The β coefficient for age (−1.248), educational level (−0.987), and household monthly income (−0.319) is negative, indicating that the higher the age, education level, and monthly income of the patient, the lower the patient’s anxiety level. The negative β coefficient for the health facility (−0.492) indicates that patients who usually visit government healthcare facilities have lower anxiety levels than patients who usually visit private healthcare facilities. The positive β coefficient for marital status—married (3.198), divorced (5.067), and widowed (1.260)—indicates that these patients have higher anxiety levels than those who are single. The highest β coefficient was observed in divorced patients, indicating that divorcees have a higher level of anxiety than married or widowed patients.

In the adjusted model, nationality and marital status (married and divorced) had an effect on anxiety scores at a significance level of 5%, while the education level and health insurance had an effect on anxiety scores at 10% significance level. The negative β coefficient for the education level (−0.918) indicates that the higher the education level, the lower the patient’s anxiety (*p*-value = 0.07). The negative β coefficient for nationality (−4020) indicates that Saudi patients have a lower level of anxiety than non-Saudi patients (*p*-value = 0.014). The positive β coefficient in married (3.107) and divorced (4.618) patients indicates that those who are married or divorced have a higher level of anxiety than those who are single. Table 4 shows that health insurance has a positive β coefficient (1.915), meaning patients who have health insurance have higher anxiety than those without health insurance.

In contrast to the results of the analysis on anxiety scores, Table 5 shows similar results between the unadjusted and adjusted models, where the number of family members, nationality, and marital status (married, divorced, and widowed) have an effect on patient depression at a significance level of 5%. Age and healthcare facility have an effect on patient depression at a significance level of 10%. The negative β coefficient for age indicates that the higher the age, the lower the level of depression. The negative β coefficient for nationality indicates that Saudi patients have lower rates of depression than non-Saudi patients (β = −3.552, *p*-value = 0.020). A negative β coefficient for healthcare facilities indicates that patients who usually visit government health facilities have lower depression levels than those who usually visit private healthcare facilities. A positive β coefficient for the number of family members indicates that a higher number of family members of patients correlates with higher levels of depression (β = 0.761, *p*-value = 0.026).

Physical activity and dietary habits showed a nonsignificant relationship with anxiety (Table A2) and depression (Table A3) in the unadjusted and adjusted models.

## 4. Discussion

In the present study, we investigated the relationship between depression, anxiety, and sociodemographic factors and lifestyle habits among patients with RA in Saudi Arabia. The findings shed light on the prevalence of depression and anxiety among patients with RA, as well as the potential factors associated with depression and anxiety. Our findings align with large cohort and cross-sectional studies from Europe, Asia, and the Middle East [9,15,16,17], which have repeatedly shown that sociodemographic vulnerabilities—particularly being female, younger, and of lower socioeconomic status—heighten the risk of psychological distress in RA.

### 4.1. Sociodemographic Factors and Depression and Anxiety

Our findings indicate that sociodemographic factors significantly influence the psychological well-being of patients with RA. Specifically, higher age, education level, and monthly income were associated with lower anxiety levels.

A large-scale retrospective cohort study that highlighted the complex relationship between lifestyle factors and the prevalence of depression and anxiety among elderly individuals reported that factors such as physical activity levels, social interaction, and dietary habits significantly influence the outcomes of depression and anxiety in this demographic [27]. However, in our study, although we could not find a significant relationship between physical activity and depression or anxiety, we found a significant impact of regular consumption of sweets and sugar on anxiety levels.

In our study, older age was associated with lower anxiety and lower depression scores. However, the literature presents mixed findings, particularly regarding depression. Some studies report that younger individuals are more prone to depressive symptoms [28,29], while others suggest that older adults have a higher likelihood of depression due to social isolation, chronic illness, or bereavement [30,31,32]. This apparent discrepancy may indicate that age exerts differential effects depending on context, culture, and support systems. In our cohort, older patients may have benefited from stronger family support networks, cultural coping mechanisms, or greater disease acceptance, which could partly explain the protective effect of age on both depression and anxiety observed in our results. Older adults may be at higher risk of depression due to several factors, including social isolation, chronic health conditions, and the psychological impact of life transitions such as retirement or the loss of loved ones [27,32,33]. A decline in physical health and reduced mobility can also contribute to feelings of helplessness and diminished self-worth, exacerbating depressive symptoms. Cognitive changes associated with aging, along with lack of access to adequate mental healthcare, further heighten the vulnerability of older individuals to depression and anxiety [27,30,32,34,35]. These factors underscore the need for comprehensive mental health interventions that address the psychological and social determinants of health in older populations. As a result, targeted intervention strategies aimed at reducing the prevalence of depression and anxiety among specific groups of older adults may be warranted.

Most patients with RA 183 (85.9%) were females, reflecting the well-established higher prevalence of RA among women. Our findings suggested that female patients were more likely to experience depression and anxiety, which aligns with previous studies reporting greater psychological vulnerability among women with RA [28,30,36]. However, another study has reported higher susceptibility among men [32]. This discrepancy may stem from differences in study populations, cultural norms, and coping mechanisms. For example, women may be more likely to report psychological symptoms, while men may underreport them due to stigma. Additionally, variations in social roles, caregiving responsibilities, and access to support systems may differentially influence mental health outcomes across genders. These contextual factors emphasize the need for gender-sensitive interventions in RA care. Although the current study’s findings on gender were not broadly generalizable, they aligned with the conclusions of Akhtar-Danesh and Landeen [8], Barua et al. [15], and Krishnaswamy et al. [36]. From a policy perspective, it may be prudent to focus more resources on addressing depression and anxiety among female patients with RA. Treatment outcomes related to fatigue improvement may also vary between genders. For instance, certain biologic therapies or interventions may have differential effects on fatigue reduction in males versus females with RA [27]. The government could implement strategies aimed at reducing stress levels in female populations with RA. For instance, regular campaigns could be organized to educate women about the potential link between stress, RA, and depression and anxiety [33].

The results of the current study corroborate prior research indicating a negative correlation between education level and depression and anxiety [32,33]. This association may be attributed to the fact that higher education enhances the individuals’ understanding of disease consequences and health improvement strategies. Consequently, the findings suggest that policymakers should focus on individuals with lower educational levels. To address this, educational programs on depression and anxiety and stress reduction could be developed and disseminated through various multilingual media, facilitating easier access to mental health information for those with less formal education.

An analysis of marital status showed that divorced participants had higher levels of anxiety (β = 5.067) than married (β = 3.198) or widowed (β = 1.260) participants. This higher anxiety among unmarried patients (never married, divorced, widowed) is consistent with the results of previous research [16,28,33,34]. High levels of depression and anxiety were statistically significant for married (*p*-value = 0.025 and *p*-value = 0.004, respectively) and divorced (*p*-value = 0.025 and *p*-value = 0.034, respectively) participants after adjusting for other sociodemographic and lifestyle factors, the use of combined DMARDs, and inflammatory markers (Table A1) [22]. This observation can be partly explained by the significant positive relationship between depression and the number of family members, where a higher number of family members is associated with increased levels of depression (β = 0.761, *p*-value = 0.026). Differences in depression and anxiety rates between Saudi and non-Saudi RA patients suggest that cultural factors, healthcare access, socioeconomic conditions, and variations in support systems (financial and medical) may contribute to the lower rates of depression and anxiety observed among Saudi patients (β = −4.020, *p*-value = 0.014 and β = −3.552, *p*-value = 0.020, respectively). Increased depression and anxiety among patients with larger families may be related to greater interpersonal conflicts, caregiver burden, financial strain, and access to public healthcare facilities.

### 4.2. Access to Healthcare and Income

The majority (84.0%) of participants had health insurance. However, after accounting for confounders, individuals with health insurance exhibited higher levels of anxiety compared to those without insurance (β = 1.915). Our findings reveal an unexpected relationship between health insurance status and anxiety levels, with insured individuals exhibiting higher levels of anxiety compared to their uninsured counterparts. This counterintuitive result challenges conventional assumptions about the psychological benefits of health insurance and warrants further investigation. Several factors may contribute to this phenomenon: increased health awareness and engagement with the healthcare system among the insured might heighten health-related concerns; the potential overutilization of medical services could lead to increased worry [16,19]; overdiagnosis appears to disproportionately affect individuals of higher socioeconomic status with comprehensive health insurance [37], as this increased access to medical services may paradoxically lead to elevated levels of anxiety and psychological distress due to a higher likelihood of encountering potentially unnecessary diagnostic procedures; and financial stress related to copayments or coverage limitations may exacerbate anxiety. We found that a substantial proportion (88.7%) of participants sought medical care from government healthcare facilities. Patients receiving care from such facilities demonstrated lower levels of depression even after adjusting for confounders. The higher depression rates observed in patients using private facilities may reflect socioeconomic stress, financial burden, or differences in accessibility and quality of care, suggesting that the type of healthcare setting could contribute to psychological distress [37]. Similar findings for this specific conclusion were not found in the literature. However, patients who regularly seek medical advice at private hospitals may face greater difficulty in scheduling free medical visits at government facilities, given the overbooking, long waiting list, and administration of medication as inpatients. Consequently, the financial burden of private hospital expenses could be stressful for these patients, potentially contributing to depression. Despite being marginally statistically insignificant (*p* = 0.067), individuals earning more than Saudi riyal 20,000 (USD 5319) per month were least susceptible to depression. This aligns with the findings of a systematic review, which indicated a correlation between financial stress and depression [19].

These findings highlight the complex interplay between health insurance, healthcare utilization, and mental well-being, highlighting the need for integrated approaches to health policy that address both physical and psychological aspects of health. Future research, including longitudinal studies and qualitative investigations, is necessary to elucidate the causal mechanisms underlying this association and to inform the development of insurance models and health education initiatives that provide coverage without inducing undue stress. Ultimately, these results emphasize the importance of a holistic approach to public health that considers the multifaceted impacts of healthcare systems on overall well-being.

### 4.3. Lifestyle Factors and Depression and Anxiety

#### 4.3.1. Diet

Our study demonstrated a statistically significant association between exclusive abstention from sugar-sweetened beverages and a reduced likelihood of exhibiting abnormal levels of anxiety (*p*-value = 0.038). This finding suggests that individuals who abstain from sugar-sweetened drinks may have a reduced susceptibility to anxiety disorders. Our investigation indicated that individuals who did not consume sugar-sweetened drinks were less likely to develop anxiety and depression. This aligns with previous studies that reported a poor quality diet may be a risk factor for depression and anxiety [38,39]. A cross-sectional study conducted in Japan [29] explored the effects of various dietary habits on depression in patients with RA. They reported increased depression levels in individuals who consumed high amounts of sugary products and concentrated fats, while those who consumed higher quantities of vegetables, fruits, and fish experienced decreased depression symptoms. A meta-analysis affirmed that the greater consumption of fish, whole grains, fruits, and vegetables is linked to a decreased risk of depression and anxiety [40,41]. Conversely, dietary patterns characterized by higher intakes of processed foods, sugary products, and saturated fats may elevate the risk of depression [42,43].

This disparity in depression risk may be attributed to the inflammatory effects of sugary products compared to the anti-inflammatory properties of fruits, vegetables, and fish. In our previous published study [22], we demonstrated that the co-occurrence of inflammation in autoimmune diseases, such as RA, and depression is a well-established phenomenon, although the precise mechanisms underlying this relationship remain incompletely understood. We proposed that one plausible explanation involves shared pathophysiological processes between peripheral immune responses and responses in the brain. This association may include the detrimental impact of pro-inflammatory cytokines on monoaminergic neurotransmission and neurotrophic factors. Importantly, the anti-inflammatory effects of certain dietary components may modulate the activity of RA, thereby influencing depression and anxiety.

#### 4.3.2. Physical Activity

In both unadjusted and adjusted models, none of the physical activity variables, including vigorous and moderate activity, showed a significant association with the anxiety or depression scores. This may be due to various sociodemographic and physiological factors, as well as the nature of the RA itself. RA and depression are bidirectionally linked, with sedentary behavior being common among patients with RA due to impaired physical function and fatigue [44]. This reciprocal relationship emphasizes a complex interplay influenced by shared inflammatory pathways, genetic factors, and psychosocial determinants. Factors like pain, fatigue, medication, diet, abnormal testosterone levels, and insufficient social support can lead to a poorer quality of life and less relief from joint symptoms, both of which contribute to reduced physical activity [45]. The hot and dry climate in many Middle Eastern countries can make outdoor exercise challenging, leading to reduced physical activity [46].

#### 4.3.3. Role of Perceived Social Support

Enhanced social support networks can reduce the psychosocial burden associated with depression and anxiety, promoting better mental health outcomes and overall well-being [47]. Policymakers can prioritize interventions that strengthen community-based support systems, such as peer support groups, counseling services, and educational programs tailored for patients with RA. These initiatives not only provide emotional and informational support but also promote resilience and coping strategies, thereby reducing the likelihood of developing depression and anxiety in individuals with RA and improving their quality of life.

Public health efforts should focus on raising awareness about the bidirectional relationship between RA, depression, and anxiety among healthcare providers, patients, and the general public. This includes promoting early screening and integrated care approaches that address physical and mental health needs. By integrating these aspects into public health policies and interventions, policymakers can effectively support individuals with RA in managing their overall health and well-being.

### 4.4. Strengths and Limitations

The study’s strengths include its relatively large sample size and comprehensive assessment of sociodemographic, clinical, and lifestyle variables. However, several limitations should be acknowledged. Firstly, the cross-sectional nature of the study limits the ability to establish causal relationships between variables. All observed relationships should therefore be interpreted as associations only, and causality cannot be inferred. Secondly, the observational design of the study precluded the measurement of disease activity scores, such as DAS-28. Data were extracted from the hospital database, and an electronic depression and anxiety questionnaire was distributed to eligible RA patients. Consequently, no face-to-face interviews were conducted to perform clinical examinations or calculate the DAS-28 score. The timing of this study, conducted after the COVID-19 pandemic, introduces potential limitations regarding the interpretation of psychological outcomes. Pandemic-related factors such as social isolation, disruptions in healthcare access, and economic stress have been shown to significantly affect mental health in patients with RA. However, because the study did not stratify participants by pre- and post-pandemic periods, it is not possible to directly assess or control for the specific impact of these pandemic-related influences on depression and anxiety levels in the sample. Additionally, the exclusion of patients receiving high-dose methotrexate (≥20 mg/week) may limit the generalizability of the findings to the broader RA population. This criterion was implemented to minimize confounding from medication-related neuropsychiatric side effects, as methotrexate at higher doses has been associated with increased risk of adverse mental health outcomes [48].

An important limitation of this study is that *p*-values were interpreted as significant without applying corrections for multiple testing. Conducting multiple statistical tests increases the risk of Type I error, meaning that some findings may be statistically significant by chance alone rather than reflecting true effects. This phenomenon, known as the multiple testing problem, can lead to inflated false-positive rates and reduce the reproducibility of results. Future studies should consider applying appropriate multiplicity correction methods, such as the Bonferroni or false discovery rate (FDR) adjustments, to control for this risk and provide more robust conclusions [49]. It is important to note that this study represents a preliminary cross-sectional investigation, laying the groundwork for a future longitudinal prospective cohort study. In this forthcoming study, patients will undergo clinical assessments, their disease activity scores will be calculated, and comparisons will be made based on the severity of their RA; we also intend to use structural equation modeling (SEM) techniques to examine both direct and indirect effects, as well as potential mediating and moderating factors within the hypothesized causal framework. However, this study provides valuable insights into the prevalence of depression and anxiety in correlation with the determinants of health among patients with RA in Saudi Arabia. The findings emphasize the importance of addressing depression and anxiety as a part of comprehensive RA management strategies. Future research should explore longitudinal associations between sociodemographic, clinical, and lifestyle factors and depression and anxiety in patients with RA to inform targeted interventions aimed at improving psychological well-being in this population.

## 5. Conclusions

In conclusion, this study highlights three key findings regarding the psychological well-being of patients with RA. First, sociodemographic factors—including age, marital status, and education—emerged as significant predictors of mental health outcomes, with adverse social determinants linked to higher risks of depression and anxiety. Second, disparities in healthcare access and income were associated with poorer psychological well-being, emphasizing the need to address socioeconomic barriers in RA management. Third, dietary habits and nutritional status were found to influence mental health, suggesting that diet is an important modifiable factor in patient care. These findings underscore two main implications: the necessity of integrated, interdisciplinary care that addresses both physical and psychological needs, and the importance of targeted interventions tailored to patients’ sociodemographic backgrounds and lifestyle factors to improve overall health outcomes. It is recommended to adopt an interdisciplinary approach that fosters collaboration among healthcare providers, including physicians, mental health specialists, nutritionists, and social workers, to effectively address the diverse needs of patients and enhance comprehensive patient care. Developing individualized care plans that consider patients’ sociodemographic backgrounds, such as cultural beliefs, economic status, and education levels, is essential to ensure that interventions are relevant and accessible. Additionally, implementing lifestyle modification programs that promote healthy changes—such as balanced nutrition, physical activity, and stress management techniques—tailored to individual circumstances can be beneficial. Strengthening community engagement and educational initiatives will raise awareness about the impact of lifestyle and sociodemographic factors on health, empowering patients to make informed health decisions. Comprehensive assessments that evaluate both physical and mental health dimensions should be incorporated into patient consultations, including screenings for depression and anxiety. Utilizing technology for continuous monitoring and feedback can further enhance patient engagement and adherence to care plans. Finally, advocating for healthcare policies and system changes that support integrative approaches will ensure that adequate resources and infrastructure are in place to implement holistic care models effectively. These strategies collectively aim to improve health outcomes by addressing both physical and mental health in a coordinated manner.

## Figures and Tables

**Table 1 ijerph-22-01625-t001:** Demographic characteristics of the study participants (n = 213) [22].

Characteristics	Subgroups	n = 213	%
Sociodemographic			
Age (years)	18–30	20	9.4
	31–40	34	16.0
	41–50	63	29.6
	51–65	96	45.1
Sex	Female	183	85.9
	Male	30	14.1
Nationality	Non-Saudi	13	6.1
	Saudi	200	93.9
Marital status	Single	29	13.6
	Married	145	68.1
	Divorced	19	8.9
	Widowed	20	9.4
Education	University degree	110	51.6
	High school degree	60	28.2
	Literate (able to read and write)	27	12.7
	Illiterate (unable to read or write)	16	7.5
Employment status	Employed	69	32.4
	Unemployed	144	67.6
Monthly household income (Saudi riyal)	0–5000	62	29.1
	5001–10,000	75	35.2
	10,001–20,000	52	24.4
	>20,000	24	11.3
Living in Riyadh	Central	17	8.0
	Eastern	42	19.7
	Western	38	17.8
	Northern	85	39.9
	Southern	31	14.6
Commute time from home to hospital	>1 h	42	19.7
	0–15 min	30	14.1
	16–30 min	88	41.3
	31–60 min	53	24.9
Usually go to government hospital	Yes	189	88.7
	No	24	11.3
Health insurance	Yes	179	84.0
	No	34	16.0
Patients receive disease-modifying antirheumatic drug (DMARD) injections at	Government health facility	65	30.5
	Private health facility	31	14.6
	Not in a health facility	117	54.9

**Table 2 ijerph-22-01625-t002:** Comparison of depression and anxiety in patients by sociodemographic characteristics.

Variable	Sub-Group	Depression				Anxiety			
		Normal	Borderline Abnormal	Abnormal	*p*-Value	Normal	Borderline Abnormal	Abnormal	*p*-Value
Gender	Male	20 (66.7%)	6 (20%)	4 (13.3%)	0.785	21 (70%)	5 (16.7%)	4 (13.3%)	0.387
	Female	110 (60.1%)	42 (23%)	31 (16.9%)		114 (62.3%)	24 (13.1%)	45 (24.6%)	
Age (years)	18–30 s	13 (65%)	5 (25%)	2 (10%)	0.320	11 (55%)	3 (15%)	6 (30%)	0.840
	31–40	18 (52.9%)	12 (35.3%)	4 (11.8%)		24 (70.6%)	3 (8.8%)	7 (20.6%)	
	41–50	35 (55.6%)	14 (22.2%)	14 (22.2%)		37 (58.7%)	11 (17.5%)	15 (23.8%)	
	51–65	64 (66.7%)	17 (17.7%)	15 (15.6%)		63 (65.6%)	12 (12.5%)	21 (21.9%)	
Education	Illiterate	8 (50%)	4 (25%)	4 (25%)	0.611	8 (50%)	2 (12.5%)	6 (37.5%)	0.339
	Literate	20 (74.1%)	5 (18.5%)	2 (7.4%)		17 (63%)	3 (11.1%)	7 (25.9%)	
	High school degree	33 (55%)	16 (26.7%)	11 (18.3%)		33 (55%)	12 (20%)	15 (25%)	
	University degree	69 (62.7%)	23 (20.9%)	18 (16.4%)		77 (70%)	12 (10.9%)	21 (19.1%)	
Employment Status	Unemployed	92 (63.9%)	32 (22.2%)	20 (13.9%)	0.307	87 (60.4%)	21 (14.6%)	36 (25%)	0.429
	Employed	38 (55.1%)	16 (23.2%)	15 (21.7%)		48 (69.6%)	8 (11.6%)	13 (18.8%)	
Household Income	0–5000	35 (56.5%)	13 (21%)	14 (22.6%)	0.067	31 (50%)	10 (16.1%)	21 (33.9%)	0.222
	5001–10,000	49 (65.3%)	15 (20%)	11 (14.7%)		52 (69.3%)	10 (13.3%)	13 (17.3%)	
	10,001–20,000	26 (50%)	18 (34.6%)	8 (15.4%)		34 (65.4%)	7 (13.5%)	11 (21.2%)	
	>20,000	20 (83.3%)	2 (8.3%)	2 (8.3%)		18 (75%)	2 (8.3%)	4 (16.7%)	
Number of Family Members	1	7 (87.5%)	1 (12.5%)	0 (0%)	0.641	7 (87.5%)	0 (0%)	1 (12.5%)	0.426
	2	7 (63.6%)	2 (18.2%)	2 (18.2%)		8 (72.7%)	0 (0%)	3 (27.3%)	
	3	8 (66.7%)	2 (16.7%)	2 (16.7%)		8 (66.7%)	2 (16.7%)	2 (16.7%)	
	4	22 (48.9%)	14 (31.1%)	9 (20%)		23 (51.1%)	7 (15.6%)	15 (33.3%)	
	>4	86 (62.8%)	29 (21.2%)	22 (16.1%)		89 (65%)	20 (14.6%)	28 (20.4%)	
Health insurance	Does not have	113 (63.1%)	40 (22.3%)	26 (14.5%)	0.191	116 (64.8%)	27 (15.1%)	36 (20.1%)	0.04 *
	Has	17 (50%)	8 (23.5%)	9 (26.5%)		19 (55.9%)	2 (5.9%)	13 (38.2%)	
Facility patient goes to	Private	10 (41.7%)	10 (41.7%)	4 (16.7%)	0.04 *	15 (62.5%)	1 (4.2%)	8 (33.3%)	0.220
	Government	120 (63.5%)	38 (20.1%)	31 (16.4%)		120 (63.5%)	28 (14.8%)	41 (21.7%)	
Living in (Riyadh)	Central	12 (70.6%)	4 (23.5%)	1 (5.9%)	0.285	11 (64.7%)	3 (17.6%)	3 (17.6%)	0.256
	Eastern	20 (47.6%)	12 (28.6%)	10 (23.8%)		23 (54.8%)	6 (14.3%)	13 (31%)	
	Northern	57 (67.1%)	15 (17.6%)	13 (15.3%)		53 (62.4%)	14 (16.5%)	18 (21.2%)	
	Southern	22 (71%)	5 (16.1%)	4 (12.9%)		26 (83.9%)	0 (0%)	5 (16.1%)	
	Western	19 (50%)	12 (31.6%)	7 (18.4%)		22 (57.9%)	6 (15.8%)	10 (26.3%)	
Commute time between home and hospital	0–15 min	20 (66.7%)	5 (16.7%)	5 (16.7%)	0.498	19 (63.3%)	5 (16.7%)	6 (20%)	0.326
	16–30 min	53 (60.2%)	25 (28.4%)	10 (11.4%)		60 (68.2%)	14 (15.9%)	14 (15.9%)	
	31–60 min	31 (58.5%)	11 (20.8%)	11 (20.8%)		30 (56.6%)	7 (13.2%)	16 (30.2%)	
	>1 h	26 (61.9%)	7 (16.7%)	9 (21.4%)		26 (61.9%)	3 (7.1%)	13 (31%)	
Received disease-modifying antirheumatic drug (DMARD) injection	Not healthcare facility	70 (59.8%)	27 (23.1%)	20 (17.1%)	0.860	78 (66.7%)	14 (12%)	25 (21.4%)	0.817
	Private facility	17 (54.8%)	8 (25.8%)	6 (19.4%)		19 (61.3%)	4 (12.9%)	8 (25.8%)	
	Government facility	43 (66.2%)	13 (20%)	9 (13.8%)		38 (58.5%)	11 (16.9%)	16 (24.6%)	

* *p*-value is statistically significant at 5% level.

**Table 3 ijerph-22-01625-t003:** Relationship between patients’ physical activities and dietary habits and depression and anxiety.

Variable	Sub-Group	Depression				Anxiety			
		Normal	Borderline Abnormal	Abnormal	*p*-Value	Normal	Borderline Abnormal	Abnormal	*p*-Value
Vigorous physical activities during 7 days	Never	110 (60.4%)	40 (22%)	32 (17.6%)	0.124	116 (63.7%)	24 (13.2%)	42 (23.1%)	0.181
	1–3 times in a week	17 (70.8%)	6 (25%)	1 (4.2%)		16 (66.7%)	5 (20.8%)	3 (12.5%)	
	4–6 times in a week	2 (50%)	0 (0%)	2 (50%)		1 (25%)	0 (0%)	3 (75%)	
	Everyday	1 (33.3%)	2 (66.7%)	0 (0%)		2 (66.7%)	0 (0%)	1 (33.3%)	
Moderate physical activities during 7 days	Never	101 (59.4%)	40 (23.5%)	29 (17.1%)	0.908	108 (63.5%)	26 (15.3%)	36 (21.2%)	0.473
	1–3 times in a week	15 (68.2%)	5 (22.7%)	2 (9.1%)		13 (59.1%)	2 (9.1%)	7 (31.8%)	
	4–6 times in a week	7 (70%)	1 (10%)	2 (20%)		5 (50%)	1 (10%)	4 (40%)	
	Everyday	7 (63.6%)	2 (18.2%)	2 (18.2%)		9 (81.8%)	0 (0%)	2 (18.2%)	
Caffeinated drinks consumed	Never	6 (54.5%)	2 (18.2%)	3 (27.3%)	0.498	7 (63.6%)	3 (27.3%)	1 (9.1%)	0.764
	1–3 times in a week	27 (64.3%)	6 (14.3%)	9 (21.4%)		26 (61.9%)	7 (16.7%)	9 (21.4%)	
	4–6 times in a week	11 (61.1%)	6 (33.3%)	1 (5.6%)		12 (66.7%)	2 (11.1%)	4 (22.2%)	
	Everyday	86 (60.6%)	34 (23.9%)	22 (15.5%)		90 (63.4%)	17 (12%)	35 (24.6%)	
Sugar-sweetened drinks consumed	Never	64 (66%)	16 (16.5%)	17 (17.5%)	0.117	66 (68%)	9 (9.3%)	22 (22.7%)	0.038 *
	1–3 times in a week	50 (64.1%)	19 (24.4%)	9 (11.5%)		54 (69.2%)	10 (12.8%)	14 (17.9%)	
	4–6 times in a week	6 (46.2%)	5 (38.5%)	2 (15.4%)		5 (38.5%)	3 (23.1%)	5 (38.5%)	
	Everyday	10 (40%)	8 (32%)	7 (28%)		10 (40%)	7 (28%)	8 (32%)	
Donuts and cakes consumed	Never	68 (63%)	21 (19.4%)	19 (17.6%)	0.604	69 (63.9%)	14 (13%)	25 (23.1%)	0.213
	1–3 times in a week	55 (56.7%)	26 (26.8%)	16 (16.5%)		61 (62.9%)	15 (15.5%)	21 (21.6%)	
	4–6 times in a week	5 (83.3%)	1 (16.7%)	0 (0%)		5 (83.3%)	0 (0%)	1 (16.7%)	
	Everyday	2 (100%)	0 (0%)	0 (0%)		0 (0%)	0 (0%)	2 (100%)	
Candy and chocolate consumed	Never	35 (72.9%)	6 (12.5%)	7 (14.6%)	0.344	33 (68.8%)	5 (10.4%)	10 (20.8%)	0.542
	1–3 times in a week	69 (57.5%)	31 (25.8%)	20 (16.7%)		78 (65%)	18 (15%)	24 (20%)	
	4–6 times in a week	12 (52.2%)	5 (21.7%)	6 (26.1%)		11 (47.8%)	4 (17.4%)	8 (34.8%)	
	Everyday	14 (63.6%)	6 (27.3%)	2 (9.1%)		13 (59.1%)	2 (9.1%)	7 (31.8%)	
Fast food consumed	Never	69 (67.6%)	17 (16.7%)	16 (15.7%)	0.49	72 (70.6%)	9 (8.8%)	21 (20.6%)	0.05 *
	1–3 times in a week	55 (55%)	28 (28%)	17 (17%)		60 (60%)	17 (17%)	23 (23%)	
	4–6 times in a week	5 (50%)	3 (30%)	2 (20%)		3 (30%)	3 (30%)	4 (40%)	
	Everyday	1 (100%)	0 (0%)	0 (0%)		0 (0%)	0 (0%)	1 (100%)	
Smoking status	Nonsmoker	125 (60.4%)	48 (23.2%)	34 (16.4%)	0.388	131 (63.3%)	28 (13.5%)	48 (23.2%)	0.923
	Smoker	5 (83.3%)	0 (0%)	1 (16.7%)		4 (66.7%)	1 (16.7%)	1 (16.7%)	

* *p*-value is statistically significant at 5% level.

**Table 4 ijerph-22-01625-t004:** Linear regression for the patients’ sociodemographic effect on their anxiety score.

Independent Variable of HADS Anxiety Score	Unadjusted Model				Adjusted Model			
	Β	*p*-Value	CI (95%)		β	*p*-Value	CI (95%)	
			Lower	Upper			Lower	Upper
(Intercept)	11.082	<0.001	6.031	7.478	8.017	0.095	−1.401	17.435
Age	−1.248 **	0.012	2.508	19.655	−0.875	0.128	−2.006	0.256
Education Level	−0.987 **	0.015	−2.253	−0.243	−0.918 *	0.078	−1.942	0.105
Monthly Household Income	−0.319 *	0.054	−1.993	0.018	−0.398	0.420	−1.372	0.576
Number of Family Members	0.380	0.497	−1.246	0.608	0.457	0.213	−0.266	1.179
Time to Hospital	0.607	0.290	−0.327	1.088	0.637	0.144	−0.221	1.495
Gender (Female)	0.904	0.150	−0.222	1.435	1.133	0.336	−1.189	3.455
Nationality (Saudi)	−4.151	0.427	−1.341	3.150	−4.020 **	0.014	−7.229	−0.810
Marital Status (Married)	3.198 **	0.010	−7.289	−1.014	3.107 **	0.025	0.401	5.813
Marital Status (Divorce)	5.067 **	0.019	0.525	5.871	4.618 **	0.025	0.595	8.640
Marital Status (Widowed)	1.260 **	0.011	1.161	8.973	1.094	0.589	−2.904	5.091
Employment Status (Employed)	−0.300	0.525	−2.647	5.166	0.027	0.977	−1.789	1.842
Health Insurance (Yes)	2.094	0.735	−2.052	1.451	1.915 *	0.082	−0.246	4.076
Health Facility (Government)	−0.492 **	0.048	0.015	4.174	−0.659	0.628	−3.345	2.027
Living in Eastern Riyadh	1.120	0.704	−3.050	2.065	0.687	0.659	−2.385	3.758
Living in Northern Riyadh	0.707	0.460	−1.867	4.107	0.200	0.890	−2.670	3.071
Living in Southern Riyadh	0.003	0.618	−2.090	3.505	−0.342	0.838	−3.651	2.967
Living in Western Riyadh	0.996	0.999	−3.205	3.211	0.732	0.661	−2.564	4.027
Receive DMARD injection by home nursing services	−0.086	0.540	−2.208	4.199	0.905	0.853	−8.728	10.537
Receive DMARD injection in a healthcare facility	0.176	0.986	−9.427	9.256	0.326	0.687	−1.270	1.923

** *p*-value is statistically significant at 5% level; * *p*-value is statistically significant at 10% level. Adjusted model: Regression analyses of sociodemographic effect on anxiety, adjusted for sociodemographic and lifestyle factors and disease activity; CI, confidence interval; DMARDs, disease-modifying antirheumatic drugs; HADS, hospital anxiety and depression scale; unadjusted model: regression analyses of sociodemographic effect on anxiety; β = standardized regression coefficient.

**Table 5 ijerph-22-01625-t005:** Linear regression for the patients’ sociodemographic effect on their depression scores.

Independent Variable of HADS Depression Score	Unadjusted Model				Adjusted Model			
	β	*p*-Value	CI (95%)		β	*p*-Value	CI (95%)	
			Lower	Upper			Lower	Upper
(Intercept)	6.585	0.103	−1.354	14.525	5.553	0.211	−3.176	14.282
Age	−0.871 *	0.066	−1.801	0.060	−0.893 *	0.094	−1.942	0.155
Educational Level	−0.034	0.943	−0.965	0.897	0.016	0.973	−0.933	0.965
Household Monthly Income	−0.587	0.179	−1.445	0.271	−0.509	0.267	−1.411	0.394
Number of Family Members	0.693 **	0.038	0.038	1.348	0.761 **	0.026	0.091	1.430
Time to Hospital	0.084	0.830	−0.684	0.851	0.179	0.657	−0.616	0.974
Gender (Female)	0.905	0.391	−1.174	2.984	1.142	0.296	−1.010	3.294
Nationality (Saudi)	−3.492 **	0.019	−6.397	−0.586	−3.552 **	0.020	−6.527	−0.577
Marital Status (Married)	3.860 **	0.002	1.385	6.335	3.740 **	0.004	1.232	6.248
Marital Status (Divorce)	4.439 **	0.017	0.821	8.056	4.037 **	0.034	0.308	7.765
Marital Status (Widowed)	4.041 **	0.029	0.423	7.658	4.374 **	0.021	0.668	8.079
Employment Status (Employed)	0.782	0.342	−0.840	2.403	1.002	0.241	−0.681	2.684
Health Insurance (Yes)	0.822	0.400	−1.104	2.747	0.623	0.540	−1.380	2.625
Health Facility (Government)	−2.213 *	0.067	−4.581	0.155	−2.426 *	0.056	−4.915	0.064
Living in Eastern Riyadh	2.122	0.132	−0.644	4.888	2.217	0.126	−0.630	5.064
Living in Northern Riyadh	1.029	0.434	−1.561	3.620	0.936	0.488	−1.725	3.596
Living in Southern Riyadh	1.244	0.409	−1.727	4.215	1.182	0.447	−1.885	4.249
Living in Western Riyadh	1.863	0.217	−1.104	4.829	1.838	0.236	−1.216	4.893
Receive DMARD injection by home nursing services	−0.329	0.940	−8.980	8.322	−0.556	0.902	−9.484	8.372
Receive DMARD injection in a healthcare facility	0.176	0.986	−9.427	9.256	0.326	0.687	−1.270	1.923

Note: ** *p*-value is statistically significant at 5% level; * *p*-value is statistically significant at 10% level. Adjusted model: Regression analyses of sociodemographic effect on depression, adjusted for sociodemographic lifestyle factors and disease activity; CI, confidence interval; DMARDs, disease-modifying antirheumatic drugs; HADS, hospital anxiety and depression scale; unadjusted model: regression analyses of sociodemographic effect on depression; β = standardized regression coefficient.

## Data Availability

The original contributions presented in the study are included in this article, its Appendix A, and our previously published manuscript [22]; further inquiries can be directed to the corresponding author.

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
