# Peer review of "Influence of Sociodemographic and Lifestyle Factors on Depression and Anxiety in Patients with Rheumatoid Arthritis in Saudi Arabia"

_ijerph, 2025, doi:10.3390/ijerph22111625_

Round 1

Reviewer 1 Report

Comments and Suggestions for Authors

Abstract and Introduction:

The text establishes RA as both a physical and psychological disease, underlining the importance of mental health alongside physical disability. References to Saudi Arabia’s mental health burden and prevalence statistics situate the study within a specific population and justify its relevance. It connects local and global evidence, citing studies from Sweden, MENA, and Saudi Arabia, giving the reader a sense of the wider research landscape. The final sentences clearly specify the study objective: examining the association between sociodemographic/lifestyle factors and depression/anxiety in Saudi RA patients. Fibromyalgia is often a disease that can be associated with RA. It would be better to mention its impact on the emotional and psychiatric conditions of patients with RA.

Phrases like “Despite the advances in pharmacological treatments targeting disease activity and pain, mental health issues remain inadequately addressed in RA management” (lines 40–42, 58–60) are repeated with little new information. This could be condensed.

The introduction sometimes jumps abruptly between general (RA globally), local (Saudi Arabia), and contextual (COVID-19) aspects. A more logical progression could be:

Burden of RA globally → Mental health comorbidities in RA → Saudi Arabia context → Gaps in research → Aim.

Some sentences are long and heavy to read. Please consider breaking them into 2 shorter ones. 

Use consistent terminology: sometimes it says “mental health issues,” other times “clinical anxiety and depression.” Aligning terms makes the text sharper.While relevant, it slightly derails the narrative.

It might work better as a brief note in the discussion section unless this study specifically stratifies patients pre- and post-COVID.

Methods: 

The section thoroughly describes study design, setting, inclusion/exclusion criteria, sample size estimation, data collection methods, and statistical analyses. The use of simple random sampling from an electronic hospital database (eSiHi) is clearly explained. Pilot testing for clarity and bilingual availability (Arabic/English) strengthens reliability. Statistical plan: The section outlines descriptive, bivariate, and regression analyses, along with adjustments for confounders.

Several sentences are overly long and could be streamlined (e.g., lines 99–107).

Inclusion/exclusion criteria: Currently presented as a dense paragraph (lines 110–117). Breaking into bullet points or clearer sentences would improve readability. The exclusion of patients on “high doses of methotrexate (≥20 mg/week)” is unusual and may need justification.

The reported “n = 213” versus “final minimum = 175 plus 20% = 213” is consistent, but the wording is a bit confusing. This could be simplified into a single, concise explanation.Regression analysis is mentioned, but the type of regression (e.g., logistic regression for categorical outcomes) is not specified. Clarifying this would improve methodological rigor.

Adjustments only for age and gender are noted — but if other confounders were considered (disease duration, comorbidities, medication use), they should be mentioned.

Results: Are adequately expressed. Tables are clear and well-realized.

The discussion connects sociodemographic, lifestyle, and healthcare factors to depression/anxiety in RA. Integration with literature: Prior studies are consistently cited, illustrating how findings align or contrast.

Again, as in other parts of the manuscripts, many sentences are overly long, with repeated qualifiers (“increased levels of depression may be related to… which may contribute to…”). Several findings are restated multiple times (e.g., education level, marital status, and diet appear in both Results and Discussion in nearly identical form). Streamlining would make the arguments sharper and easier to follow. Contradictions around age: In Results, older patients had lower anxiety (β = –1.248), but in Discussion, older adults are presented as at higher risk of depression. This needs clarification (distinguish between anxiety vs depression effects of age). Gender discussion cites conflicting evidence; a more nuanced synthesis (e.g., “our findings align with X but differ from Y; this discrepancy may stem from…”) would improve balance.

The conclusion is strong but a bit too broad and repetitive. It could be shortened to emphasize the three main findings (sociodemographic predictors, healthcare access, diet) and the two main implications (integrated care & targeted interventions).

Comments on the Quality of English Language

Some of the sentences are too long and may be hard-readable. Breaking them up into 2 sentences would make the article more readable and better understood.

Reviewer 2 Report

Comments and Suggestions for Authors

The manuscript addresses an important gap by exploring the influence of sociodemographic and lifestyle factors on depression and anxiety among RA patients. The study design and analysis are generally sound, but there are some issues with clarity, statistical rigor, and interpretation that require attention before publication.

1. While the cross-sectional design is appropriate for exploratory associations, causality cannot be inferred. This limitation should be emphasized more strongly in the discussion. 

2. P-values are interpreted as "significant" without corrections for multiple testing, which may inflate Type I error. Consider addressing this limitation.

3. Adjustment for disease activity (e.g., CRP, ESR) is mentioned, but these are only briefly reported. A more detailed explanation of how disease activity was controlled in the regression would strengthen the conclusions.

4. Figures illustrating key data would improve readability.

5. The finding that patients using private facilities had higher depression rates is intriguing. Could this be due to socioeconomic stress, cost burden, or differences in quality of care? More interpretation is needed.

6. In Table 1, there appear to be extra numbers included in some rows and columns (for example, numbering next to the variable names). These numbers may not be part of the data itself and could reduce readability.

Reviewer 3 Report

Comments and Suggestions for Authors

The manuscript addresses an important question regarding mental health in RA patients, with regional relevance to Saudi Arabia. The sample size is adequate, the HADS instrument is validated, and the focus on sociodemographic and lifestyle factors adds value. However, there are some issues that require major revision before the manuscript can be considered for publication.

  1. Reference [19] Misattribution
    • The Methods and Results sections repeatedly cite reference [19] as though it describes the same Saudi RA cohort and prior analyses. However, reference [19] in fact relates to a Latin American COVID-19 population, not RA patients in Saudi Arabia.
    • This is a serious miscitation that misleads the reader about the source of methodology and results. The authors must clarify:
      • Has this Saudi RA dataset been published previously?
      • If yes, provide the correct reference and specify what is new in this manuscript.
      • If no, explicitly state that this dataset is original and unpublished.
  2. Sample Size Inconsistency
    • At line 99, the study is described as including 210 RA patients, while at line 119 the sample size is calculated as n = 213. Results tables also report n = 213.
    • The discrepancy between 210 and 213 must be corrected and made consistent across the Abstract, Methods, Results, and Tables. It should be clear whether 210 patients completed the survey or 213 were analyzed.
  3. Cross-Sectional Limitations
    • The design does not allow causal inference. Some sentences (e.g., “consuming sugar-sweetened drinks was significantly associated with anxiety”) imply causality and should be rephrased to emphasize association only.
  4. Statistical Reporting
    • Tables are very dense and difficult to read. Consider moving detailed dietary/physical activity subgroup analyses to supplementary material.
    • Report exact p-values consistently. Interpret borderline findings (0.05–0.1) with caution.
    • Ensure regression tables include clear reference categories for categorical variables.
  5. Results–Discussion Consistency
    • In the Results, older age is negatively associated with anxiety, while the Discussion suggests older adults may be more prone to depression. These interpretations should be reconciled so that readers are not left confused.
  6. Integration with Literature
    • The Discussion cites some international studies but would benefit from broader comparison with global RA literature (Europe, Asia, North America). How do the prevalence rates of anxiety and depression in this Saudi cohort compare with international averages?

Round 2

Reviewer 2 Report

Comments and Suggestions for Authors

The authors addressed all the issues raised by this reviewer.